# Investigating the Effects of Exogenous and Endogenous 2-Arachidonoylglycerol on Retinal CB1 Cannabinoid Receptors and Reactive Microglia in Naive and Diseased Retina

**DOI:** 10.3390/ijms242115689

**Published:** 2023-10-28

**Authors:** Sofia Papadogkonaki, Dimitris Spyridakos, Emmanouela Lapokonstantaki, Nikos Chaniotakis, Alexandros Makriyannis, Michael S. Malamas, Kyriaki Thermos

**Affiliations:** 1Department of Pharmacology, School of Medicine, University of Crete, Heraklion, 71003 Crete, Greece; sofia.papadogkonaki@unige.ch (S.P.); medp2011897@med.uoc.gr (D.S.); 2Department of Chemistry, University of Crete, Heraklion, 71003 Crete, Greece; chemp1134@edu.chemistry.uoc.gr (E.L.); chaniotakis@uoc.gr (N.C.); 3Center for Drug Discovery and Departments of Chemistry and Chemical Biology and Pharmaceutical Sciences, Northeastern University, Boston, MA 02115, USA; a.makriyannis@northeastern.edu (A.M.); malamas.michael@gmail.com (M.S.M.)

**Keywords:** 2-arachidonoylglycerol, CB1 and CB2 cannabinoid receptors, MAGL, ABHD6, neuroinflammation, microglia, neuroprotection, eye drops

## Abstract

The endocannabinoid system (ECS) is a new target for the development of retinal disease therapeutics, whose pathophysiology involves neurodegeneration and neuroinflammation. The endocannabinoid 2-arachidonoylglycerol (2-AG) affects neurons and microglia by activating CB1/CB2 cannabinoid receptors (Rs). The aim of this study was to investigate the effects of 2-AG on the CB1R expression/downregulation and retinal neurons/reactive microglia, when administered repeatedly (4 d), in three different paradigms. These involved the 2-AG exogenous administration (a) intraperitoneally (i.p.) and (b) topically and (c) by enhancing the 2-AG endogenous levels via the inhibition (AM11920, i.p.) of its metabolic enzymes (MAGL/ABHD6). Sprague Dawley rats were treated as mentioned above in the presence or absence of CB1/CB2R antagonists and the excitatory amino acid, α-amino-3-hydroxy-5-methyl-4-isoxazolepropionic acid (AMPA). Immunohistochemistry, Western blot and a 2-AG level analyses were performed. The 2-AG repeated treatment (i.p.) induced the CB1R downregulation, abolishing its neuroprotective actions. However, 2-AG attenuated the AMPA-induced activation of microglia via the CB2R, as concurred by the AM630 antagonist effect. Topically administered 2-AG was efficacious as a neuroprotectant/antiapoptotic and anti-inflammatory agent. AM11920 increased the 2-AG levels providing neuroprotection against excitotoxicity and reduced microglial activation without affecting the CB1R expression. Our findings show that 2-AG, in the three paradigms studied, displays differential pharmacological profiles in terms of the downregulation of the CB1R and neuroprotection. All treatments, however, attenuated the activation of microglia via the CB2R activation, supporting the anti-inflammatory role of 2-AG in the retina.

## 1. Introduction

A fully functional endocannabinoid system (ECS) is present in the vertebrate retina and is comprised of the G protein-coupled cannabinoid CB1 and CB2 receptors (Rs), the endocannabinoids, anandamide (AEA) [1] and 2-arachidonoylglycerol (2-AG) [2,3] and the enzymes responsible for their synthesis and metabolism [4]. Previously, 2-AG was shown to be the most abundant endogenous cannabinoid compared to anandamide in the brain [5] and retina [6,7].

Moreover, 2-AG activates both the CB1R and CB2R originally shown to be located in the CNS (brain/retina) and in inflammatory conditions, respectively. The CB1R is located in the outer plexiform layer (OPL), the inner nuclear and inner plexiform layers (INL/IPL) and the ganglion cell layer (GCL) of the rodent retina [6,8,9,10]. The CB2R is primarily localized in the retinal glial cells [11,12]. However, its mRNA and protein expression have been reported in the retinal layers (GCL, IPL, INL, OPL) and inner segments of the photoreceptors of the rat [10,13,14] and mouse retina [15,16,17]. 

Furthermore, 2-AG undergoes rapid degradation, and therefore its physiological actions are hampered. Three serine hydrolases are responsible for 2-AG metabolism: monoacylglycerol lipase (MAGL), responsible for approx. 85% of 2-AG hydrolysis in the brain [18,19,20], and the secondary enzymes, (α)/β-hydrolase domain-containing6 (ABHD6) and 12 (ABHD12), responsible for approx. 4% and 9%, respectively, of the total 2-AG hydrolysis [18]. 

In the brain, the MAGL is expressed presynaptically, as is the CB1R [21]. MAGL immunoreactivity (IR) was reported in the mouse retina (GCL, IPL, OPL) and nerve fiber layer (NFL) [11,22,23] and in Müller and amacrine cells, a subpopulation of cone bipolar cells in the INL, as well as in the IPL and GCL, in the rat retina [24]. Marrs et al. [25] reported that ABHD6, in the brain, is expressed in postsynaptic structures, neurons (cell soma and dendrites), BV-2 cells and less in microglia. However, ABHD6 was also shown to be located on microglia/macrophages and colocalized with the CB2R [26]. In the mouse retina, the ABHD6 expression was reported in the IPL, INL and GCL and at the dendrites of ganglion or displaced amacrine cells [22].

The endocannabinoid and synthetic cannabinoid activation of the CB1R and CB2R, when administered acutely, provided neuroprotective and anti-inflammatory effects, respectively, in the brain [27,28] and retina [9,17,29,30,31,32]. Blockade of endocannabinoid metabolism, by inhibiting the activity of their hydrolytic enzymes, increases the endogenous levels of AEA and 2-AG and thus presents an indirect route of enhancing their signaling activity via cannabinoid receptor activation [33]. Considering the vast pleiotropic properties mediated via the ECS, many investigations have identified the neuroprotective actions of endocannabinoid metabolic inhibitors in various brain neurodegenerative disease models [34,35,36,37]. More specifically, MAGL inhibition has been shown to display neuroprotective properties in Parkinson’s [38], Alzheimer’s [39] and Huntington’s diseases [40], epilepsy [41] and others. However, fewer studies have examined the neuroprotective properties of endocannabinoid metabolic enzyme inhibitors in the retina [29,31].

The neuroprotective actions of cannabinoids are compromised under chronic agonist treatment due to the downregulation of the CB1R and the subsequent loss of its downstream signaling cascade, leading to the development of tolerance [42,43,44]. We initially addressed this issue in the retina by investigating the effects of the subchronic or chronic exogenous administration (i.p.) of AEA and synthetic cannabinoids, MethAEA and HU210. Our results indicated that these treatments led to the downregulation of the CB1R and the attenuation of the downstream prosurvival actions of phosphatidylinositol 3 kinase/protein kinase B (PI3K/Akt) and/or mitogen-activated protein kinase/extracellular-signal-regulated kinase (MEK/ERK1/2) signaling pathways [45]. 

Earlier brain studies reported that chronic pharmacological inhibition or deletion of MAGL caused the desensitization of the CB1R and behavioral tolerance [46,47]. However, it was subsequently shown that repeated low-dose administration of a MAGL inhibitor maintains the beneficial antinociceptive and anti-inflammatory effects, without producing functional CB1R tolerance or cannabinoid dependence, in an experimental model of neuropathic pain [48]. There are no data reported to date regarding the effects of chronic administration of 2-AG metabolic enzyme inhibitors in the retina.

In the present study, we investigated the role of 2-AG in (a) the CB1R expression, signaling and neuroprotection and (b) the regulation of reactive microglia in the model of AMPA excitotoxicity. For this purpose, we employed three paradigms, involving repeated intraperitoneal (i.p.) treatment with 2-AG or AM11920 (MAGL/ABHD6 inhibitor), as well as repeated 2-AG topical (eye drop) administration. Our findings suggest that repeated 2-AG treatments in the three paradigms lead to differential pharmacological profiles with respect to the downregulation of the CB1R and neuronal protection. However, all treatments attenuated the activation of microglia via the CB2R activation, supporting the anti-inflammatory role of 2-AG in the retina. A schematic representation of the 2-AG signaling pathways on retinal neurons and microglia is presented in Appendix A.

## 2. Results

### 2.1. Dose-Dependent Effect of Repeated Intraperitoneal 2-AG Administration on CB1 Receptor Expression

The effect of the endogenous cannabinoid 2-AG in the naive rat retina, after repeated treatment, was investigated. In this study, 2-AG (25, 50, 100 μg/kg, i.p.) was administered daily for 4 d, and immunohistochemical studies were performed to assess its effect on the expression of the CB1 receptor in the retina (Figure 1A). A quantitative analysis of the immunohistochemical data showed that 2-AG induced a dose-dependent reduction in the CB1R expression (Figure 1B). Administered at a dose of 50 μg/kg and 100 μg/kg (i.p., *n* = 5), 2-AG attenuated the CB1R-IR by 65% and 70%, respectively, compared to the control animals (*** *p* < 0.001). The lower dose of 25 μg/kg (i.p., *n* = 4) had no effect on the CB1R expression compared to the controls (*p* > 0.05). A Western blot analysis was also performed in order to confirm the 2-AG-induced CB1R downregulation. The dose of 50 μg/kg (i.p.) was chosen for this study, according to the immunohistochemical data showing that there was no significant difference in the efficacy between the 50 μg/kg and 100 μg/kg doses (Figure 1B). In fact, 2-AG (50 μg/kg, i.p., *n* = 5, * *p* < 0.05) induced in a statistically significant manner a 26% reduction in the CB1R protein expression after the 4 d treatment compared to the controls (*n* = 3) (Figure 1C). These findings suggest that repeated 2-AG administration (i.p.) induces the downregulation of the CB1R in the rat retina.

### 2.2. Effect of Repeated Intraperitoneal Administration of 2-AG on Retinal Neuroprotection and CB1R Expression in the Model of AMPA Excitotoxicity

In order to assess the neuroprotective properties of the repeated 2-AG treatment, the model of α-amino-3-hydroxy-5-methyl-4-isoxazolepropionic acid (AMPA) excitotoxicity was employed according to Papadogkonaki et al. [45]. Intravitreal administration of AMPA induced in a statistically significant manner a 55% reduction in NOS^+^ amacrine cells compared to the controls (*n* = 6, **** *p* < 0.0001, Figure 2A,B). The repeated administration of 2-AG (50 μg/kg, i.p., 4 d) did not reverse the AMPA-induced attenuation of the number of NOS^+^-positive amacrine cells (*n* = 6, **** *p* < 0.0001 compared to the controls, *p* > 0.05 compared to the AMPA group) suggesting that 2-AG did not provide neuroprotection to this type of amacrine cells (Figure 2B). The lack of neuroprotection was correlated with the ability of 2-AG to induce a 56% attenuation of CB1R expression (downregulation) (*n* = 5, ** *p* < 0.01 compared to the controls, * *p* < 0.05 compared to the AMPA group) (Figure 2C). AMPA did not affect the CB1R-IR (*n* = 4, *p* > 0.05) compared to the controls (*n* = 5, Figure 2C).

We conclude from these data, that the repeated 2-AG treatment (i.p.) did not reverse the AMPA-induced attenuation of bNOS^+^ amacrine cells, and thus it did not provide neuroprotection due to the downregulation of the CB1R.

### 2.3. Impact of CB1 and CB2 Inhibitors on the Effect of the Intraperitoneal Repeated Administration of 2-AG on AMPA-Induced Microglial Cell Activation

The excitatory amino acid AMPA induced a fourfold increase in the number of reactive microglial cells (Iba1^+^) four days after administration (**** *p* < 0.0001, *n* = 5) compared to the controls (Figure 3A,B). The administration of 2-AG (50 μg/kg, i.p.) reversed this effect by attenuating the number of reactive microglia to the control levels (** *p* < 0.01, *n* = 5) compared to the AMPA group. To assess the involvement of the CB1R or CB2R in the actions of 2-AG, antagonists of both receptors were employed. AM630 (CB2R antagonist) reversed the 2-AG effects in microglia (*n* = 5, * *p* < 0.05 compared to the AMPA + 2-AG group, *** *p* < 0.001 compared to the controls). This hypothesis was further supported by the observation that the CB1 receptor antagonist SR141716 did not reverse the 2-AG effect (*n* = 5, ** *p* < 0.01 compared to the AM630 group, *** *p* < 0.001compared to the AMPA group (Figure 3B)). These data suggest that repeated 2-AG administration (i.p.) leads to the attenuation of the number of reactive microglia via the activation of the CB2R. Therefore, the CB2Rs located on microglia are also involved in the actions of 2-AG in the retina.

### 2.4. Effect of Repeated 2-AG Intraperitoneal Administration on CB1R Downstream Signaling

Previous studies performed in our laboratory reported that the prosurvival pathway, PI3K/Akt is involved in the neuroprotective actions of 2-AG according to Kokona et al. [31]. ERK1/2 was also involved in the neuroprotective actions of AEA and MethAEA [9].

In this study, we employed a Western blot analysis using a monoclonal antibody that recognizes isoforms Akt 1 phosphorylated at the Ser473 residue. However, this antibody might also detect Akt2 and Akt3 isoforms when phosphorylated at the same residue. We also employed a monoclonal antibody that detects ERK1 (p44 MAP kinase) and ERK2 (p42 MAP kinase) when they are phosphorylated dually at Thr202/Tyr204 and Thr185/Tyr187, respectively. We examined the effect of the repeated administration of 2-AG (50 μg/kg, i.p.) on the CB1R-induced phosphorylation of PI3K/Akt and ERK1/2 kinases (Figure 4A). In this study, 2-AG (50 μg/kg, i.p., *n* = 5) increased the phosphorylation levels of the Akt protein 0.3-fold (* *p* < 0.05 compared to the controls, *n* = 3, Figure 4B). Moreover, 2-AG (*n* = 5) did not alter the phosphorylation levels of the ERK1/2 kinases (*p* > 0.05 compared to the controls, *n* = 4) (Figure 4C). These findings suggest that the CB1R downregulation does not affect the 2-AG signaling activity via the PI3K/Akt pathway.

### 2.5. Levels of 2-AG in Rat Retina after Repeated Administration of Exogenous 2-AG and MAGL/ABHD6 Inhibitor AM11920

A high-resolution mass spectrometry–liquid chromatography–mass spectrometry(HR-LC-MS/MS) analysis revealed that the endogenous 2-AG levels in the control animals (*n* = 4) were 3.250 ± 0.987 ng/mg. The repeated exogenous 2-AG administration (*n* = 5, 50 μg/kg, i.p.) raised significantly the 2-AG levels in the retina (5.353 ± 0.503 ng/mg, * *p* < 0.05 compared to the controls). An even greater increase in the endogenous 2-AG levels (7.142 ± 1.119 ng/mg) was observed in the retinas of the animals that received repeatedly the ABHD6/MAGL inhibitor AM11920 (*n* = 5, 500 μg/kg, i.p., *** *p* < 0.001 compared to the controls, * *p* < 0.05 compared to the 2-AG group, Figure 5). These data support that both the exogenous 2-AG and AM11920 (endogenous 2-AG) reach the retina and are responsible for the pharmacological actions of 2-AG mediated by the CB1 and CB2 receptors.

### 2.6. Effect of Repeated Treatment with MAGL/ABHD6 Inhibitor on Expression of CB1 Receptor and Retinal Neuroprotection

The blockade of the 2-AG hydrolytic enzymes (MAGL/ABHD6) with the repeated AM11920 (dual inhibitor) treatment did not affect the expression of the CB1R at the doses studied (250, 500 μg/kg, i.p., 4 d, *n* = 4, *p* > 0.05 compared to the controls, Figure 6A), and the CB1R was not affected by the repeated AMPA or AMPA + AM11920 treatment (AMPA (*n* = 4, *p* > 0.05) or AM11920 (*n* = 4, *p* > 0.05) compared to the controls (*n* = 5), Figure 6B). As shown in Figure 6C, the administration of AMPA reduced the number of NOS^+^ amacrine cells by 48% (*n* = 4, **** *p* < 0.0001 compared to the controls). The repeated AM11920 treatment restored the number of amacrine cells (**** *p* < 0.0001 compared to the AMPA group, *n* = 4) to the control levels (^ns^ *p* > 0.05 compared to the controls). These data suggest that the AM11920-induced increase in the endogenous 2-AG levels does not lead to the downregulation of the CB1R and thus provides neuroprotection to retinal amacrine cells.

### 2.7. Effect of AM11920 on the AMPA-Induced Increase in the Number of Activated Microglia

The effect of AM11920 on the AMPA-induced activation of microglia was examined. As shown in Figure 6D, AMPA induced a fivefold increase in the number of reactive microglial cells (*n* = 4, *** *p* < 0.001 compared to the controls). AM11920 (*n* = 3, 500 μg/kg, i.p., 4 d) reversed the number to the control levels (* *p* < 0.05 compared to the AMPA group (*n* = 4), ^ns^ *p* > 0.05 compared to the controls (*n* = 4)). The attenuation of the AMPA-induced increase in reactive microglia by AM11920 was due to the 2-AG-mediated activation of the CB2R, as indicated by the results shown in Figure 3B.

### 2.8. Effect of Topical Administration of 2-AG on CB1R Expression and Neuroprotection

The repeated administration of 2-AG (1%, 4 d, *n* = 3) provided partial neuroprotection. It reversed the AMPA-induced attenuation of bNOS-IR with a 1.7-fold increase (*n* = 4, ** *p* < 0.01 compared to the control or AMPA groups, Figure 7A). AMPA and 2-AG (1%, 4 d) had no effect on the CB1R expression (*p* > 0.05 compared to the controls (*n* = 3) or the AMPA group (*n* = 3), Figure 7B).

In the same model, a Western blot analysis revealed that AMPA decreased the protein levels of the antiapoptotic Bcl-2 by 41% (*n* = 4, * *p* < 0.05 compared to the controls). The topical treatment with 2-AG (1%, 4 d) reversed the AMPA-induced effect on the Bcl-2 expression to the control levels (100%) (*n* = 4, * *p* < 0.05 compared to the AMPA group, Figure 7C,D).

Moreover, the AMPA administration was shown to activate the cell death SAPK/JNK pathway. It increased the phosphorylation levels of SAPKs/JNKs 0.5-fold (*n* = 3, * *p* < 0.05 compared to the controls (*n* = 4), Figure 7C,E). The repeated topical administration of 2-AG (1%, 4 d) decreased phospho-SAPK/JNK to the control levels (*n* = 4, ** *p* < 0.01 compared to the AMPA group, ^ns^ *p* > 0.05 compared to the controls). These data suggest for the first time that 2-AG administered as drops protects the retina against the AMPA excitotoxicity by increasing the Bcl-2 expression and attenuating the SAPK/JNK activation.

### 2.9. Effects of Repeated Topical Administration of 2-AG on Iba-1^+^ Microglial Cells in the AMPA Model of Excitotoxicity

The potential anti-inflammatory effects of topical 2-AG administration against the AMPA-induced microglial activation were also examined (Figure 8A). The intravitreal administration of AMPA induced a ninefold increase in the number of reactive microglial cells (*n* = 4, **** *p* < 0.0001) compared to the controls (*n* = 5) (Figure 8B). The eye drop administration of 2-AG (1%, 4 d) partially reduced the AMPA-induced microglial activation by 18% (*n* = 3, * *p* < 0.05 compared to the AMPA group, **** *p* < 0.0001 compared to the controls (Figure 8B). Topical administration of 2-AG may be beneficial as a therapeutic for retinal neuroinflammatory diseases.

## 3. Discussion

The present study provides new evidence regarding the effects of 2-AG on the CB1R expression and reactive microglia in the control and diseased retinas when administered (i.p. or topically) repeatedly, either exogenously or via the administration of the MAGL/ABHD6 inhibitor and the subsequent increase in the endogenous 2-AG levels. Our findings suggest that 2-AG, in the three paradigms employed, leads to differential pharmacological profiles in terms of the downregulation of the CB1R and neuroprotection. However, all treatments attenuated the activation of microglia via the CB2R activation, supporting the anti-inflammatory role of 2-AG in the retina.

### 3.1. Repeated Intraperitoneal 2-AG Treatment

Kokona et al. [31] reported that the acute intravitreal administration of the endocannabinoid 2-AG provides neuroprotection to NOS-expressing retinal amacrine cells via the activation of both the CB1R and CB2R, as was confirmed by the actions of 2-AG in the presence of the CB1R and CB2R antagonists. In the present study, the repeated 2-AG administration led to the attenuation of the protein expression of the CB1R (downregulation) and the abolishment of neuroprotection of the above-mentioned amacrine cells (Figure 2). These findings are in agreement with previous data suggesting that AEA, MethAEA and HU-210, the cannabinoids known to protect the retina against the AMPA excitotoxicity, when administered acutely [9], do not display the same response under subchronic or chronic conditions [45].

In a previous study, we showed that the synthetic CB1R/CB2R agonist WIN55,212-2 via the activation of the CB2R reversed the AMPA-induced increase in reactive microglia [49]. Here, we show that repeated administration of 2-AG also reversed the increase in the AMPA-induced reactive microglia via the activation of the CB2R, as confirmed by the reversal of the 2-AG actions by the CB2R antagonist AM630, but not the CB1R antagonist SR141716. These findings imply that the CB2R in microglia is not affected (downregulated) by repeated 2-AG treatment and suggest that 2-AG via the CB2R plays an important role in regulating retinal microglia responses to inflammation. To support this tenet further, one could study the CB2R expression in this paradigm. However, due to the methodological limitations of the available CB2R antibodies that are related to their reduced specificity [50,51,52], one runs the risk of obtaining deceiving data. Our present findings regarding the role of the CB2R are in agreement with other reports that indicate the putative therapeutic benefits of the CB2R agonists for various retinopathies whose pathophysiology involves neuroinflammation [16,53].

Acute treatment with 2-AG has been shown to activate the CB1R and its downstream prosurvival PI3K/Akt pathway [31], suggesting its role as a neuroprotectant of retinal neurons. The interaction of 2-AG with the Akt signaling may be due to the CB1R-dependent “biased signaling”, which pertains to the ability of different cannabinoid ligands to activate divergent intracellular signaling pathways [54]. Laprairie et al. [55] performed an extensive analysis of the functional selectivity profile of different endogenous and synthetic CB1R ligands on striatal medium spiny neurons in vitro. They discovered that the endocannabinoids 2-AG and AEA displayed a stronger biased signaling towards the Gα_i/o_- and Gβγ-mediated Akt activation compared to the synthetic cannabinoid ligands [55]. In the present study, the CB1R downregulation induced by the repeated treatment with 2-AG did not lessen, but increased, the Akt phosphorylation in contrast to the attenuation observed for other CB1R agonists [45]. In agreement with our data, repeated treatment with the ABHD6 inhibitor WWL70 (3 and 7 days) has been shown to increase the Akt phosphorylation in a mouse model of TBI injury [56].

A plausible explanation for such an increase in the ph-Akt expression may be due to the 2-AG-induced activation of the CB2R. As previously mentioned, the CB2R actions on microglial activation indirectly imply that the repeated treatment with 2-AG, at the same dose (50 μg/kg, i.p.) that induced the CB1R downregulation, did not affect the CB2R expression. In addition to this, inflammation and neurodegeneration are known to promote an increased expression of CB2Rs in reactive microglia in the brain [57,58,59] and retina disease models [12]. The CB2R-mediated activation of the PI3K/Akt pathway has been observed in reactive microglia [54] and found to be essential for shifting the microglial polarization from the pro-inflammatory M1 towards the anti-inflammatory M2 phenotype [60]. A functional role of the CB2R-mediated activation of the PI3K/Akt pathway in providing neuronal protection has also been confirmed under neurodegenerative conditions in the brain [61,62].

The CB1R activation also induces the phosphorylation of the prosurvival intracellular kinase, ERK1/2. AEA administered acutely increased the ERK1/2 phosphorylation [9] but had no effect when administered subchronically [45]. In the present study, the repeated treatment with 2-AG had no effect on the ERK1/2 phosphorylation in agreement with the findings related to the subchronic AEA treatment [45].

### 3.2. Blockade of 2-AG Hydrolytic Enzymes: Endogenous 2-AG Actions

The 2-AG metabolic enzyme inhibitors are considered important therapeutic targets for CNS brain disease [35]. However, there are very few studies in the literature pertaining to their role in the retina. Acute treatment with the novel inhibitors of the 2-AG hydrolytic enzymes, AM12100 (ABHD6) and AM11920 (dual MAGL/ABHD6) [63] attenuated the AMPA-induced microglial/macroglial activation (Iba1-IR/GFAP-IR) and produced dose-dependent partial neuroprotection, with the dual inhibitor being more efficacious [31]. These results suggest that the pharmacological blockade of the 2-AG hydrolytic enzymes and the subsequent increase in the endogenous 2-AG levels may provide a beneficial therapeutic strategy for neurodegenerative retinopathies.

In the present study, we showed for the first time that the dual MAGL/ABHD6 inhibitor, AM11920 (500 μg/kg), when administered repeatedly, has no effect on the CB1R expression and displays neuroprotection of NOS-expressing amacrine cells against AMPA excitotoxicity. At the same dose, AM11920 attenuated the AMPA-induced increase in the number of reactive microglia. Long et al. [64] were the first to establish the selectivity profile of the MAGL inhibitor JZL184 in vivo, in the brain, and report that its acute treatment elevated the 2-AG levels fivefold. However, chronic treatment with JZL184 (40 mg/kg, i.p., once daily, 6 d) was reported to inhibit the MAGL activity and elevate the brain 2-AG levels up to 10-fold over the control group, leading to tolerance [47]. Soon after, Kinsey et al. [48] reported that the tolerance of the CB1R, induced by JZL184, is dose-dependent. Low-dose administration of JZL184 (less than 8 mg/kg, 6 d) did not lead to the CB1R tolerance and maintained the pharmacological effects of 2-AG in the brain. In agreement with the latter, chronic JZL184 (10 mg/kg, 14d) treatment also increased the 2-AG levels in the hippocampus without inducing the desensitization of the CB1R [65].

In the present study, we employed HR-LC/MS/MS to analyze the 2-AG levels in the retinas of the animals that received 2-AG exogenously, or their 2-AG levels were induced endogenously by the dual ABHD6/MAGL inhibitor. The animals that received repeated exogenous 2-AG treatment for 4 d displayed a marked increase in the 2-AG levels, compared to the control animals, while a higher increase in the 2-AG endogenous levels was observed with the repeated AM11920 treatment (Figure 5). We conclude that the repeated treatment with AM11920 at a dose of 500 μg/kg employed in this study does not affect the expression of the CB1R and does not lead to its downregulation in agreement with the dosing regiments mentioned above [48]. The 2-AG levels measured in the retinas after its repeated exogenous treatment were lower than those induced by AM11920. Yet, the repeated 2-AG treatment led to the downregulation of the receptor. A possible explanation could be that the exogenously administered 2-AG initially induced a greater increase in its retinal levels, sufficient to affect the CB1R expression, but soon after, it was rapidly hydrolyzed by the endogenous metabolic enzymes MAGL/ABHD6, resulting in lower 2-AG detected levels after 24 h. In contrast, the initial elevation of the 2-AG levels induced by AM11920 did not affect the expression and function of the CB1R. As a result, AM11920 blocked the 2-AG hydrolysis and allowed its detection with HR-LC/MS/MS. The AM11920 dose (500 μg/kg) we selected falls under the lower-limit dose known not to induce the CB1R tolerance according to Kinsey et al. [48].

### 3.3. Topically Administered 2-AG

Endocannabinoids due to their lipophilic nature possess the ability to reach the BRB when administered topically as eye drops [30,53,66]. Taking into account the chronic nature of retinal diseases and the noninvasive character of eye drop administration; a growing body of investigations underlines the advantage of this route of administration for the treatment of retinal diseases [67,68,69,70].

In this study, we report for the first time that topical administration of 2-AG (1%, 4 d), as eye drops, presented antiapoptotic and anti-inflammatory properties. Intravitreal injection of AMPA increased the phosphorylation of the c-Jun N-terminal kinases (JNKs) in the rat retina. JNKs, also known as stress-activated kinases (SAPKs), belong to the MAPK family, and among their various physiological actions is the initiation of apoptotic cell cascades [71,72], while they also play a crucial role in regulating retinal apoptosis [73]. Under excitotoxic conditions, an increased phosphorylation of JNKs is observed, which contributes to the subsequent neuronal cell death [74,75,76]. Here, the topical application of 2-AG via eye drops reduced the phosphorylation of the JNKs and reversed the AMPA-induced reduction in the antiapoptotic Bcl-2 protein. In compliance with our results, brain studies indicate that the antiapoptotic properties of cannabinoids are based on their ability to inhibit the phosphorylation of the JNKs [77] and restore the expression of Bcl-2 [78,79]. It has been shown that the binding of Akt1, one of the downstream signaling pathways of 2-AG, to the JNK-interacting protein 1 (JIP1) inhibits the potentiation of the JNK activity mediated by the JIP1 and thus attenuates excitotoxic apoptosis in the brain [73].

The antiapoptotic properties of the 2-AG repeated dosing as eye drops are due to the lack of the retinal CB1R downregulation. However, it should be noted that despite the fact that eye drops represent a less invasive and more patient-friendly route of ocular drug delivery, its major drawback is the low bioavailability due to the low corneal retention time and several other barriers the drug must cross in order to reach the retina [80]. Based on this fact, we can speculate that in the eye drop treatment, a lesser amount of 2-AG reached the retina, efficient enough to provide neuroprotection but not enough to induce the CB1R downregulation. This is further supported by the fact that topical administration of 2-AG only partially reduced the number of reactive microglial cells, in contrast to the intraperitoneal administration, where full suppression of microglial reactivity was noted. Therefore, it becomes apparent that we cannot compare the results of the 2-AG treatments (4 d; topical vs. intraperitoneal), since more studies (time and dosing) are needed in order to elucidate the properties of 2-AG under subchronic/chronic treatment in the eye drop route of administration. However, we can conclude that 2-AG administered as eye drops protects the retina against cell death via the CB1R activation and attenuates the number of reactive microglial cells responsible for cytokine release and neuroinflammation via the activation of the CB2R [58].

## 4. Materials and Methods

### 4.1. Animals

Adult male and female Sprague Dawley rats (200–300 g) were employed in compliance with EU Directive 2010/63/EU for animal experiments, the Greek National legislation (Animal Act., P.D. 160/91) and the 3R principle (replacement, reduction, refinement). Animals were housed 2 to 3 in cages of appropriate size according to their sex and maintained on a 12 h light/dark cycle at room temperature (22 ± 2 °C) with food and water ad libitum. Euthanization was performed with CO_2_ inhalation. All protocols were approved by the Animal Care Committee assigned by the local Veterinarian Authorities and by the Ethics Committee of the University of Crete (project authorization protocol code 109015/30-05-2017, protocol code 207608/18-09-2020).

### 4.2. Drugs and Treatment

#### 4.2.1. Drugs

The endogenous cannabinoid 2-arachidonoylglycerol (2-AG) (Cayman Chemicals, Ann Arbor, MI, USA) was dissolved in a vehicle solution of absolute ethanol/water for injection (2% EtOH/98% WFI). AM11920 (dual MAGL/ABHD6 inhibitor), SR141716 (CB1R antagonist) and AM630 (CB2R antagonist), kindly provided by Dr. M.S. Malamas and Prof. A. Makriyiannis),were dissolved in a vehicle solution of dimethyl sulfoxide (Sigma Aldrich, St. Louis, MO, USA)/water for injection (8% DMSO/92% WFI). All the above agents were prepared as stock solutions (10^−2^ M), stored at −20 °C and further diluted prior to treatment to appropriate dose/concentration with WFI for injections (i.p., final injection volume: 500 μL). (R,S)-a-Amino-3-hydroxy-5-methyl-4-isoxazolepropionic acid hydrobromide (AMPA, Tocris, Bristol, UK) was initially dissolved in WFI and further diluted in 50 mM phosphate-buffered saline (PBS) for intravitreal administration (final injection volume: 5 μL/eye). For the eye drop administration experiment, 2-AG was diluted in a vehicle solution of absolute ethanol (Honeywell, Muschegon, MI, USA)/PBS (50 mM, final administration volume: 20 μL).

#### 4.2.2. Intraperitoneal Treatment

Naive rats received 2-AG (25, 50 or 100 μg/kg, i.p.) or AM11920 (250, 500 μg/kg, i.p.) daily for 4 days (repeated treatment). Control rats were administered with vehicle solution [81].

#### 4.2.3. Topical Treatment

Topically, 2-AG was administered via eye drops for 4 consecutive days (once daily). Rats were immobilized, and their upper eyelid was gently pulled in order to facilitate the administration. A total volume of 20 μL of 2-AG (1%, diluted in 50 mM PBS) or PBS (50 mM, control animals) was administered in two to three drops to each eye using a pipette.

#### 4.2.4. In Vivo AMPA-Induced Model of Excitotoxicity and Treatment

The AMPA excitotoxicity model was employed as described by Kiagiadaki and Thermos [82]. Animals were anesthetized with xylazine (10 mg/kg, i.p.) and ketamine (100 mg/kg, i.p.) and placed in a stereotaxic apparatus to stabilize the head and facilitate intravitreal administration. Rat eyes received intravitreally 5 μL PBS (50 mM) or 5 μL AMPA (8.4 mM, diluted in PBS, Tocris, Bristol, UK) [9]. Twenty-four hours later, 2-AG (50 μg/kg, i.p., 4 d) or AM11920 (500 μg/kg, i.p., 4 d) or 2-AG (1%, eye drops, 4 d) were administered. A subgroup of animals that received 2-AG (50 μg/kg, i.p.) were pre-treated with either SR141716 (CB1R antagonist, 500 μg/kg, i.p., 4 d) or AM630 (CB2R antagonist, 500 μg/kg, i.p., 4 d) 30 min prior to the 2-AG administration. Control rats were administered with a vehicle solution. All experimental paradigms and treatments are presented in Appendix A.

### 4.3. Immunohistochemical Studies

#### 4.3.1. Tissue Preparation

Twenty-four hours (24 h) after the last treatment, animals were euthanatized, and eyecups (posterior part of the eye, consisting of the retina, RPE, choroid and sclera) were, collected for immunohistochemical studies. After removal, the rat eyes were fixed by immersion in 4% paraformaldehyde (PFA, (Sigma Aldrich, St. Louis, MO, USA)) in 0.1 M phosphate buffer (PB) for 45 min at 4 °C. Subsequently, the anterior part of the eyeballs (cornea, lens and vitreous humor) was removed, and the eyecups were further fixed in 4% PFA in 0.1 M PB for 1.5 h at 4 °C. When isolated, the eyecups were incubated in 30% sucrose overnight at 4 °C for cryoprotection. The tissues were frozen in isopentane at −45 °C for 1 min, using O.C.T. compound (optimal cutting temperature compound) and kept at −80 °C until further use.

The eyecups were sectioned vertically near the optic nerve head using a cryostat (Superfrost, Erie Scientific, Portsmouth, NH) at −25 °C and stored at −20 °C. Serial vertical sections of 10 μm each were spread into 4 gelatin-covered slides (6 sections per slide) so that each slide contained a representative part of the central retina, including the optic nerve head. Retinal samples for Western blot and HR-LC/MS/MS analyses were prepared by rapid isolation of retina from the eyes and storage at −80 °C until further use.

#### 4.3.2. Immunohistochemistry

Immunohistochemistry was performed to assess the immunoreactivity (IR) of the CB1R and specific retinal markers to evaluate cell loss and neuroprotection. Cryostat sections were washed twice with 0.1 M Tris-HCl (Sigma Aldrich, St. Louis, MO, USA) buffer(TBS, pH 7.4 (10 min each wash)), incubated with blocking buffer (0.1 M TBS) containing 3.3% normal goat serum (NGS) for 30 min to block the nonspecific binding sites, washed three times in 0.1 M TBS (5 min each wash) and subsequently incubated with the primary antibodies diluted in 0.1 M TBS containing 0.3% Triton X-100 and 0.5% NGS overnight at room temperature (RT). The sections were washed three times with 0.1 M TBS (5 min each) and incubated for 1.5 h in dark conditions with the appropriate secondary antibody. Finally, the sections were washed as previously, incubated for 2 min with DAPI (Sigma-Aldrich, St. Louis, MO, USA) diluted in 0.1 M TBS (1:1000) and cover-slipped with mounting medium. Negative controls were obtained by omitting the primary antibody.

The following primary antibodies were employed for immunohistochemistry: anti-CB1R (rabbit polyclonal, 1:300, Cayman Chemicals, Ann Arbor, MI, USA) to assess CB1R expression, anti-brain nitric oxide synthetase (bNOS), a marker for nitric oxide synthetase (NOS) expressing amacrine cells (rabbit polyclonal, 1:2000, Sigma-Aldrich, St. Louis, MO, USA), and anti-Iba-1 (rabbit polyclonal, 1:2.500, WAKO Chemicals, Osaka, Japan), a marker of microglia. The secondary antibody CF543 goat anti-rabbit IgG (H + L) (1:1000, Biotium, Fremont, CA, USA) was employed for all the primary antibodies listed above. Representative photomicrograph of retinal tissue stained only with the secondary antibody CF543 goat anti-rabbit is shown in Appendix A as the negative control for the immunofluorescence images.

### 4.4. Microscopy and Quantification Studies

Light microscopy images were obtained using a Leica DMLB fluorescence microscope (HC PL Fluotar, 40×/0.70 lens, Leica Microsystems, Wetzler, Germany) with a Leica DC 300F camera. Light adjustments of immunofluorescence were set using the Leica software before image acquisition and remained fixed. Two images were taken from 3 slices (2 images/slice) of each retina (6 images per retina) near the optic nerve head (central retina) containing the OPL, INL, IPL and GCL. The appropriate areas were delineated according to the expression of each antibody using the public-domain software ImageJ 1.44. Antibody immunostaining was calculated by measuring the mean grey value per area in each image, and the mean of the six values was used for each sample (retina).

bNOS: bNOS^+^ retinal neurons were counted manually along the entire retinal tissue (3 slices/retina) in the INL (NOS expressing amacrine cells) and GCL (displaced amacrine cells).

CB1R: The GCL was delineated for quantification of CB1R-IR in retinal sections in naive retinas due to the robust CB1R expression on ganglion cells (Papadogkonaki et al., 2019). CB1R expression was also detected in the following retinal layers, INL, IPL and GCL, as mentioned in the introduction. Therefore, in sections used for the AMPA excitotoxicity model, the area containing the INL, IPL and GCL was delineated in order to also quantify CB1-IR in NOS^+^ amacrine cell somata and processes.

Iba-1: Iba-1positive (Iba-1^+^) microglial cells were divided according to their morphology as reactive or resting [49]. Double staining with DAPI enabled the discrimination of Iba-1^+^ cells and artifacts. Iba-1^+^ reactive cells were manually counted in the area containing INL, IPL, GCL, and the ratio of Iba-1^+^ reactive cells/area was calculated. Adobe Photoshop ver. 7.0 software (Adobe Systems, San Jose, CA, USA) was used to crop the images and adjust brightness and contrast. Measurements of mean gray value and manual counting of cells were performed prior to any adjustment with Adobe Photoshop.

### 4.5. Western Blot Analysis

Homogenization of retinal samples was performed by sonication in a NP-40 lysis buffer containing 50 mM Tris-HCl pH 7.5, 150 mM NaCl, 1% NP-40, 0.1% DOC, 0.1 mM PMSF, a protease inhibitor (Sigma-Aldrich, St. Louis, MO, USA) and phosphatase inhibitor (Roche, Mannheim, Germany). The samples were centrifuged at 10,000× *g* for 20 min at 4 °C. Total protein was extracted by centrifugation and resuspended in SDS loading buffer containing 0.1 M DTT (DiThioThreitol, Sigma-Aldrich, St. Louis, MO, USA).

Retinal lysates were loaded (100 μg of total protein/well) [83] into a 12.5 or 15% acrylamide (Sigma-Aldrich, St. Louis, MO, USA) gel, according to the molecular weight of the protein of interest, separated by SDS-PAGE and then transferred onto nitrocellulose membranes (Macherey-Nagel, Duren, Germany). Membranes were washed for 3 × 10 min with 1× TBS/Tween20 (TBST, Sigma-Aldrich, St. Louis, MO, USA) buffer and then blocked with 3% BSA (Sigma-Aldrich, St. Louis, MO, USA) in TBST for 1 h at RT under agitation. 

Antibodies against CB1R (rabbit polyclonal 1:500, Cayman Chemicals, Ann Arbor, MI, USA), the phosphorylated or total isoforms of Akt, ERK1/2 and SAPK/JNK proteins, Bcl-2 or GAPDH (rabbit monoclonal antibodies, 1:1000, Cell Signaling Technology, Danvers, MA, USA) were diluted in blocking buffer, and membranes were incubated with the appropriate antibody overnight at 4 °C under agitation. Subsequently, membranes were washed 3 × 10 min with TBST and incubated with anti-rabbit-HRP (1:5000, Cell Signaling Technology, Danvers, MA, USA) or anti-mouse-HRP (1:10,000, Sigma-Aldrich, St. Louis, MO, USA) for 1 h at RT under agitation. Membranes were incubated with a Thermo Scientific Pierce ECL detection kit (ThermoFischer Scientific, Waltham, MA, USA) for the visualization of the bands, and the optical density of the bands in each blot was quantified using the ImageJ [version 1.44, NIH and Laboratory for Optical and Computational Instrumentation (LOCI), University of Wisconsin, Madison, WI, USA] software.

### 4.6. Detection of Retinal 2-AG Levels with HR-LC/MS/MS

#### 4.6.1. Sample Preparation

Pre-weighted retinal samples, from animals that were treated with 2-AG (50 μg/kg, i.p., 4 d) or AM11920 (500 μg/kg, i.p., 4 d), were put on ice. Extraction of 2-AG was performed by adding 500 μL of ice-cold acetonitrile in each sample (1 sample = one retina per animal). Samples were then sonicated on ice for a total duration of 30 s, separated into 5 s intervals with 5 s pause in between, in order to avoid sample overheating. Homogenized samples were subsequently centrifuged at 13,500 rpm for 1 h at 4 °C; the supernatant was collected and stored at −20 °C until the HR-LC-MS/MS analysis.

#### 4.6.2. HR-LC-MS/MS Parameters

Qualitative and quantitative determination of 2-AG was performed on a QExactive™ Plus Quadrupole-Orbitrap™ mass spectrometer (Thermo Scientific, Waltham, MA, USA) and the Ultimate 3000 HR-LC System (ThermoFischer Scientific, Waltham, MA, USA). A Hypersil Gold separation column (100 mm × 4.6 mm i.d., 5 μm) was used at 50 °C. Solvents A (H_2_O, 0.1% formic acid) and B (Acetonitrile, ACN, 0.1% formic acid, Thermo Fischer Scientific, Waltham, MA, USA) were used. The separation was performed at a flow rate of 0.4 mL/min by gradient elution, starting at 20% B for 3 min, which increased linearly to 100% B within 1 min. After holding these conditions for 3 min, the initial conditions were restored for 3 min before the next analysis. 

The ESI-MS/MS detection was performed in positive ion mode, and the monitoring conditions were optimized for the target compound. The conditions were as follows: spray voltage was set at 4.2 kV, and the capillary temperature was held at 300 °C. Sheath gas and auxiliary gas pressures were set at 38 and 11 Arb, respectively. The ions used to quantify 2-AG and for the calibration curve were 2,872,375 as the quantization ion and 2,692,271 as the confirmation ion, with the NCE set at 15. The calibration curve was constructed for the concentrations of 10, 25, 50, 100, 250 ng/mL. All parameters for PRM transitions were optimized in order to obtain the highest sensitivity.

### 4.7. Statistical Analysis

Quantification data were expressed as the mean ± SD (standard deviation) and analyzed using GraphPad Prism 9.0 (GraphPad Software, Inc., San Diego, CA, USA). Two-tailed unpaired Student’s *t*-test was used for the statistical analysis of two groups. One-way ANOVA followed by Tukey’s post hoc analysis was used for multiple group comparisons. Differences between the groups were considered statistically significant when *p* < 0.05.

## 5. Conclusions

To summarize, the present findings show for the first time that repetitive 2-AG (i.p.) treatment for four consecutive days leads to the downregulation of the CB1R and to its inability to provide neuroprotection to retinal neurons. 

However, an increase in the endogenous 2-AG levels produced via the blockade of the MAGL/ABHD6 hydrolytic enzymes does not affect the CB1R expression and function in the retina. More studies are essential in order to elucidate the cutoff dose of the 2-AG inhibitor (AM11920) under which the CB1R signaling activity cannot be compromised. This, in combination with the CB2R-mediated anti-inflammatory properties will support the recruitment of AM11920 as a new putative therapeutic for retina and brain neurodegenerative diseases.

The repeated 2-AG eye drop administration was shown to be efficacious as a neuroprotectant in the diseased retina via mechanisms that may involve the interplay of Akt/JNK/Bcl-2. Further studies are needed in order to confirm this hypothesis, which in conjunction with the 2-AG influence on microglial physiology may prove that eye drop administration is beneficial for the treatment of neurodegenerative and neuroinflammatory retinal diseases.

## Figures and Tables

**Figure 1 ijms-24-15689-f001:**
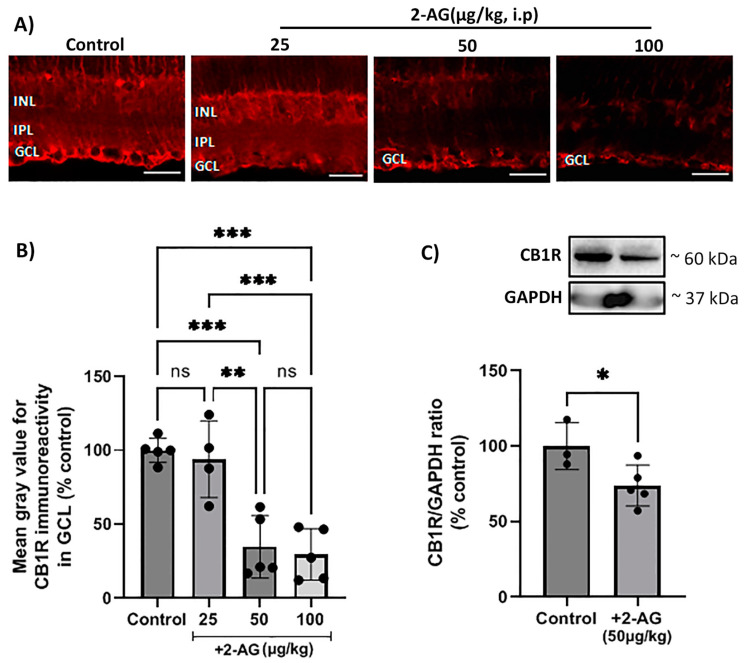
Effect of repeated 2-AG (i.p.) treatment on CB1R expression. (**A**) Representative images of CB1R-IR (red) in retinas of control and 2-AG-treated (25, 50, 100 μg/kg, i.p., 4 d) rats. Magnification: ×40. Scale bar: 50 μm. (**B**) Quantitative analysis of CB1R-IR in the ganglion cell layer (GCL) (** *p* < 0.01, *** *p* < 0.001). (**C**) Western blot and quantitative data showing CB1R expression. Data are expressed as the mean ± SD (*n* = 3–5); * *p* < 0.05, ^ns^ *p* > 0.05 (ns = no significance).

**Figure 2 ijms-24-15689-f002:**
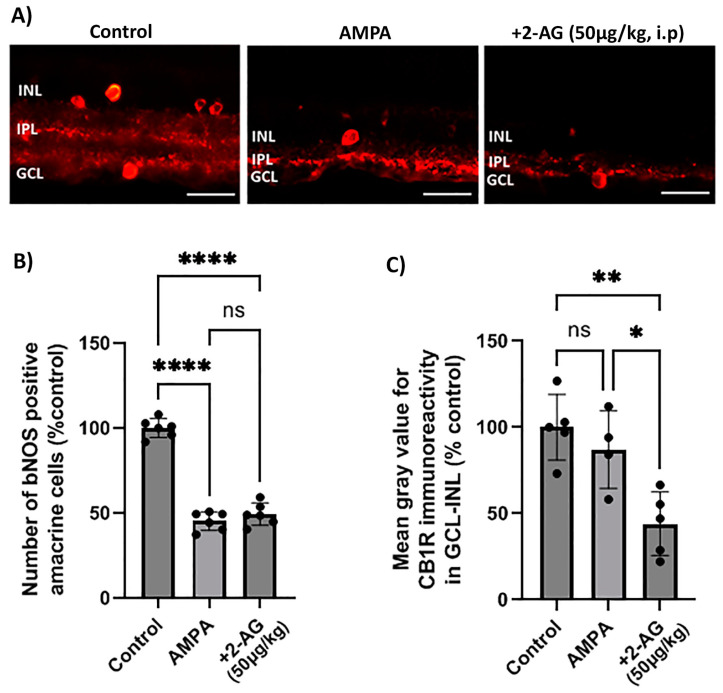
Effect of repeated administration of 2-AG (i.p.) on bNOS- and CB1R-IR in the model of α-amino-3-hydroxy-5-methyl-4-isoxazolepropionic acid (AMPA) excitotoxicity. (**A**) Representative images of bNOS-IR (red) in retinas of control rats and rats treated with AMPA (8.4 mM, I.V.T.) and AMPA + 2-AG (50 μg/kg, i.p., 4 d). Magnification: ×40. Scale bar: 50 μm. (**B**) Quantitative analysis data of bNOS-positive amacrine cells. (**C**) Quantification study of CB1R-IR in the area containing INL, IPL and GCL. Data are expressed as the mean ± SD (*n* = 4–6); ^ns^ *p* > 0.5, * *p* < 0.05, ** *p* < 0.01, **** *p* < 0.0001 (ns = no significance).

**Figure 3 ijms-24-15689-f003:**
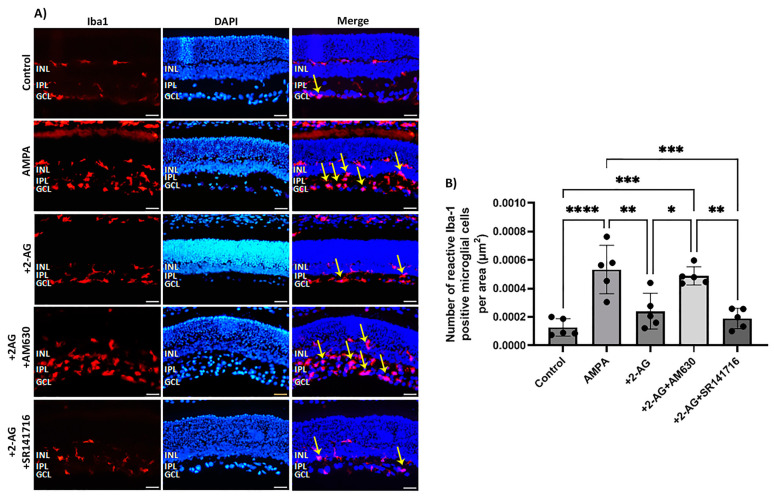
Effect of 2-AG on AMPA-induced microglial cell activation. (**A**) Representative images of Iba-1 (red) and DAPI (blue) IHC staining showing positive microglial cells in retinas of control rats and rats treated with AMPA (8.4 mM, I.V.T.), AMPA + 2-AG (50 μg/kg, i.p.), AMPA + 2-AG + AM630 (500 μg/kg, i.p.) or AMPA + 2-AG + SR141716 (500 μg/kg, i.p.). Reactive microglial cells are depicted by the yellow arrows. Magnification: ×20. Scale bar: 50 μm. (**B**) Quantitative analysis of reactive Iba-1^+^ microglial cells. Data are expressed as the mean ± SD (*n* = 5); * *p* < 0.05, ** *p* < 0.01, *** *p* < 0.001, *****p* < 0.0001.

**Figure 4 ijms-24-15689-f004:**
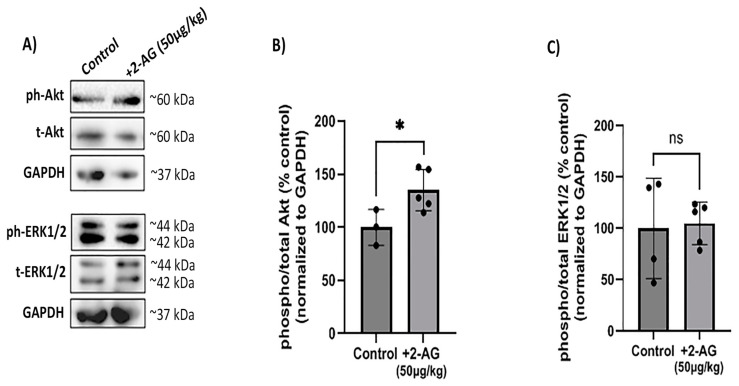
Effect of repeated 2-AG treatment on CB1R downstream signaling. (**A**) Western blot analysis of phosphorylated (phospho-) or total isoforms of Akt and ERK1/2 kinases. (**B**,**C**) Quantitative data showing the effect of 2-AG (50 μg/kg, i.p., 4 d) on Akt and ERK1/2 phosphorylation. Data are expressed as the mean ± SD (*n* = 3–5); ^ns^ *p* > 0.05, * *p* < 0.05 compared to the controls (ns = no significance). Phospho- or total Akt isoforms were observed in one single band of ~60 kDa, whereas phospho- or total isoforms of ERK1/2 kinases were observed as two distinct bands of ~42 and ~44 kDa.

**Figure 5 ijms-24-15689-f005:**
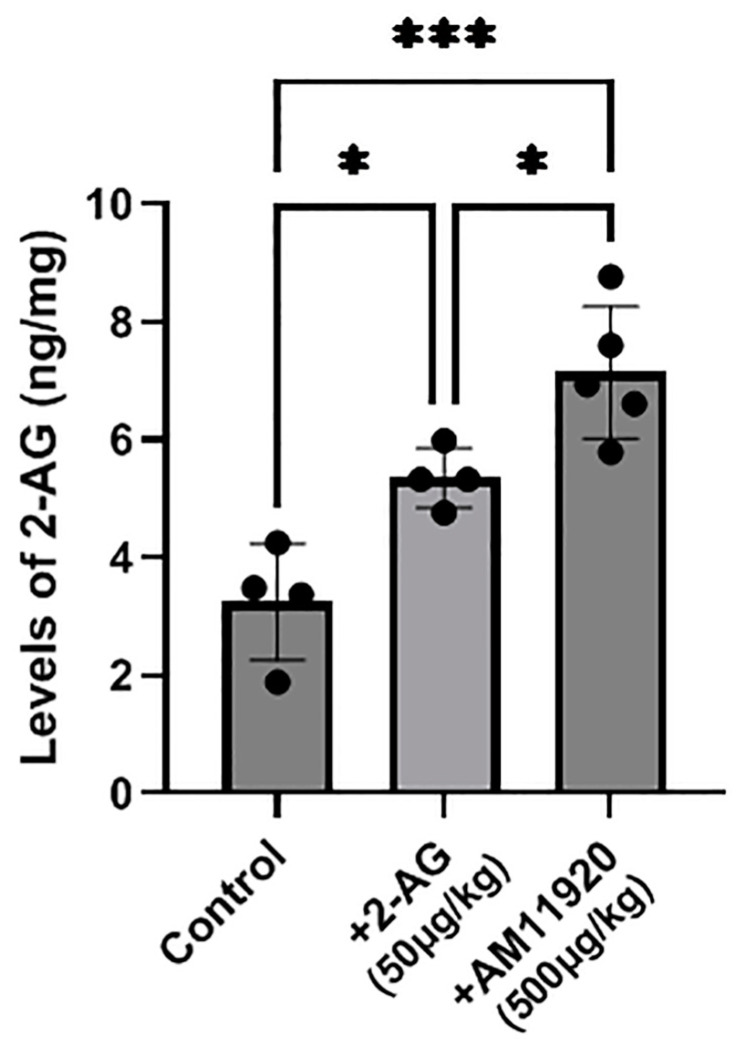
Measurement of 2-AG levels by HR-LC-MS/MS in retinas of control animals and animals treated with 2-AG (50 μg/kg, i.p., 4 d) or AM11920 (500 μg/kg, i.p., 4 d). Data are expressed as the mean ± SD (*n* = 4–5); * *p* < 0.05, *** *p* < 0.001.

**Figure 6 ijms-24-15689-f006:**
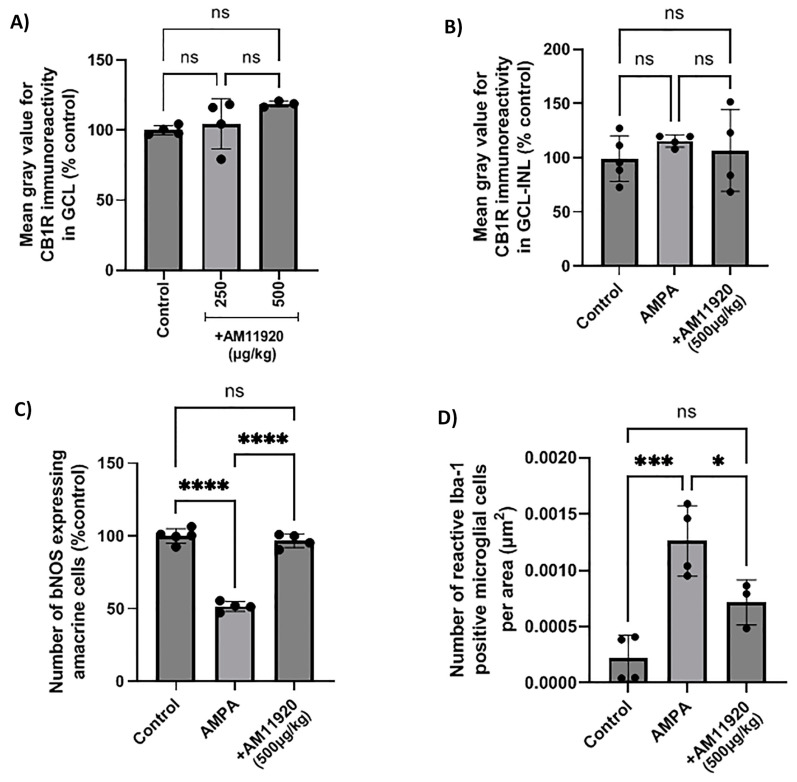
Effect of the MAGL/ABHD6 inhibitor, AM11920, on CB1 receptor expression and on retinal neuroprotection in naϊve rat retina or in the model of AMPA excitotoxicity. (**A**) Quantification data of CB1R-IR in the retina of control rats and rats treated with AM11920 (250, 500 μg/kg, i.p., 4 d). Quantitative analysis of CB1R-IR (**B**), bNOS-IR (**C**) and Iba-1-IR (**D**) in control rats and rats treated with AMPA (8.4 mM, I.V.T.) or AMPA + AM11920 (500 μg/kg, i.p., 4 d). Data are expressed as the mean ± SD (*n* = 3–5); ^ns^ *p* > 0.05, * *p* < 0.05, *** *p* < 0.001, **** *p* < 0.0001 (ns = no significance).

**Figure 7 ijms-24-15689-f007:**
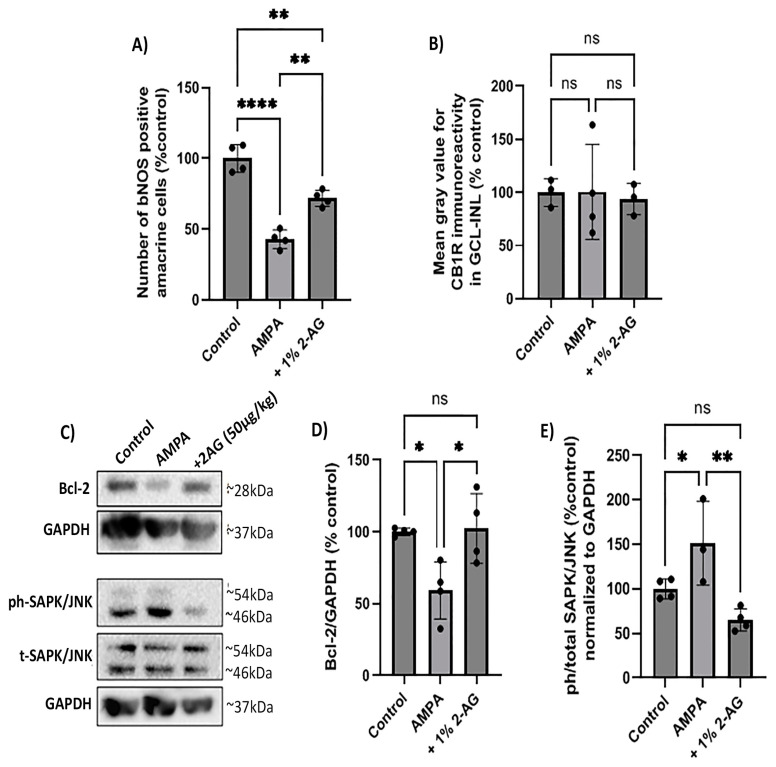
Neuroprotective effect of repeated topical 2-AG administration in the model of AMPA excitotoxicity. (**A**) Quantification of the number of bNOS^+^ amacrine cells and (**B**) CB1R-IR in control retinas and retinas treated with AMPA (8.4 mM, I.V.T.) and AMPA + 2-AG (1%, eye drops, 4 d). (**C**) Western blot analysis of Bcl-2 and the phosphorylated (phospho-) or total isoforms of SAPKs/JNKs. (**D**) Quantitative data showing the effect of topical treatment with 2-AG on Bcl-2 expression and (**E**) SAPK/JNK phosphorylation. Data are expressed as the mean ± SD; ^ns^ *p* > 0.05, * *p* < 0.05, ** *p*< 0.01, **** *p* < 0.0001 compared to the controls; *p* < 0.01 compared to the AMPA group (ns = no significance). Bcl-2 was detected as a single band of ~28 kDa, whereas phospho- and total SAPK/JNK isoforms were observed as two distinct bands of ~46 and ~54 kDa.

**Figure 8 ijms-24-15689-f008:**
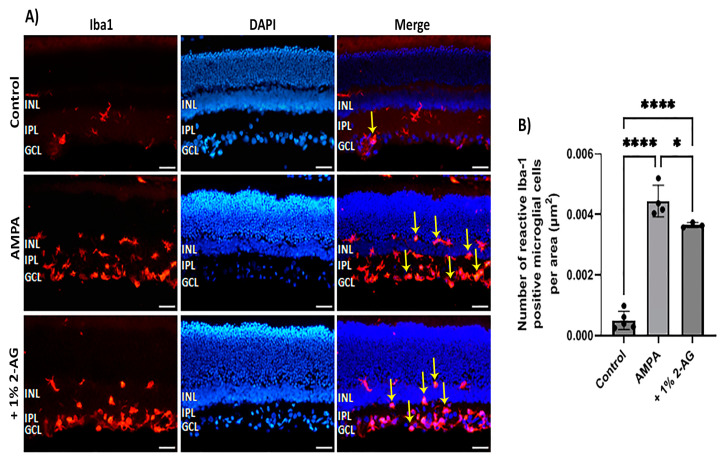
Effects of topical administration of 2-AG on AMPA excitotoxicity-induced microglial activation. (**A**) Representative images of Iba-1 (red) and DAPI (blue) IHC staining in the retina of control rats and rats treated with AMPA (8.4 mM, I.V.T.) and AMPA + 2-AG (1%, eye drops, 4 d). Yellow arrows depict reactive microglial cells. Yellow arrows depict reactive microglial cells. Magnification: ×20. Scale bar: 50 μm. (**B**) Quantitative analysis of the number of reactive Iba-1^+^ microglial cells in the area containing INL, IPL and GCL. Data are expressed as the mean ± SD (*n* = 3–5); * *p* < 0.05, **** *p* < 0.0001.

## Data Availability

Not applicable.

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
