# Peer review of "Investigating the Effects of Exogenous and Endogenous 2-Arachidonoylglycerol on Retinal CB1 Cannabinoid Receptors and Reactive Microglia in Naive and Diseased Retina"

_ijms, 2023, doi:10.3390/ijms242115689_

Round 1
Reviewer 1 Report
Comments and Suggestions for Authors
ijms-2670262, Investigating the role of exogenous and endogenous 2-arachidonoylglycerol on retinal CB1 cannabinoid receptors and reactive microglia in naïve and diseased retina
The work presented here seems to be conducted correctly and the results were properly analyzed and presented. There are some issues that could be corrected in order to improve the quality of the paper.
Row 41 and elsewhere in the manuscript: present each abbreviation, even if it was presented in the abstract. The main part of the paper and the abstract should be considered as to distinct sections. The same goes for the captations of the figures or the tables.
Row 75: the authors should better details the neuroprotective effects of drugs targeting the endocannabinoid system and provide specific examples. See for example, the following articles:
Targeting Monoacylglycerol Lipase in Pursuit of Therapies for Neurological and Neurodegenerative Diseases, Molecules, 2021
Targeting Endocannabinoid Signaling: FAAH and MAG Lipase Inhibitors, Annual Review of Pharmacology and Toxicology, 2021
Row 91: prepare a scheme to present all the pathways described in the introduction section
Section 2.1. the same idea is repeated 3 times “no significant difference in the efficacy between the 50μg/kg and 100μg/kg dose”
Row 127, explain what is AMPA
Row 146: “AMPA induced an increase in the number of reactive microglia cells”. Present the dose and a quantitative measure, like 3 or 4 folds increase. The same for “2-AG (50μg/kg, i.p) reversed this effect”. Did it reverse the effect completely (100%)? Or only by 50 or 25%? This approach should be used for all the other assays performed in this study.
Row 150, present the dose of AM630 that produced the effect
Row 171, it is not clear if the assay measured all isoforms of Akt and which position is phosphoryled (ex Thr308 or Ser473). The same for ERK1 and 2.
Row 297, there are several studies on the effect of 2AG on Akt function. Here is just one example: Type 1 cannabinoid receptor ligands display functional selectivity in a cell culture model of striatal medium spiny projection neurons. The authors should better discuss the previous data in context with their research.
There are several editing mistakes. The authors should carefully verify the paper.
Comments on the Quality of English LanguageThe quality of English is OK
Author Response
Manuscript ID: ijms-2670262
We thank the reviewer for his/her comments.
We have revised the manuscript according to your recommendations and also responded as to why some proposed experiments were not performed.
Response to Reviewer 1
Comment 1: Row 41 and elsewhere in the manuscript: present each abbreviation, even if it was presented in the abstract. The main part of the paper and the abstract should be considered as to distinct sections. The same goes for the captions of the figures or the tables.
Response: Abbreviation of 2-arachidonoylglycerol was added: (2-AG). Row 41. The text and captions/tables were checked for similar mistakes.
Comment 2: Row 75: the authors should better details the neuroprotective effects of drugs targeting the endocannabinoid system and provide specific examples. See for example, the following articles:Targeting Monoacylglycerol Lipase in Pursuit of Therapies for Neurological and Neurodegenerative Diseases, Molecules, 2021Targeting Endocannabinoid Signaling: FAAH and MAG Lipase Inhibitors, Annual Review of Pharmacology and Toxicology, 2021.
Response: The following text was included in paragraph 5 of the introduction. “Considering the vast pleiotropic properties mediated via the ECS, many investigations have identified the neuroprotective actions of endocannabinoid metabolic inhibitors in various brain neurodegenerative disease models (Zanfirescu et al., 2021; van Egmond et al., 2021) [36,37]. More specifically, MAGL inhibition has been shown to display neuroprotective properties in Parkinson’s (Mounsey et al.[38],), Alzheimer’s (Hashem et al. [39]) and Huntington’s disease (Roperh et al. [40]), epilepsy (Terrone et al[41]) and others. However, fewer studies have examined the neuroprotective properties of endocannabinoid metabolic enzyme inhibitors in the retina [29,31]”.
Comment 3: Row 91: prepare a scheme to present all the pathways described in the introduction section
Response: A schematic representation of 2-AG’s signaling pathways on retinal neurons and microglia is presented in Figure S1 at the last paragraph of the introduction.
Comment 4: Section 2.1. the same idea is repeated 3 times “no significant difference in the efficacy between the 50μg/kg and 100μg/kg dose”.
Response: The statement “no significant difference in the efficacy between the 50μg/kg and 100μg/kg dose” is now shown only once in the whole paragraph.
Comment 5: Row 127, explain what is AMPA
Response: α-amino-3-hydroxy-5-methyl-4-isoxazolepropionic acid (AMPA) is an excitatory amino acid that mimics the effects of the neurotransmitter glutamate. Both AMPA and glutamate bind to AMPA receptors and induce excitotoxicity.
The full name of AMPA was included in the following sentence in the abstract, “Sprague-Dawley rats were treated as mentioned above in the presence or absence of CB1/CB2R antagonists and the excitatory amino acid, α-amino-3-hydroxy-5-methyl-4-iso-xazolepropionic acid (AMPA)” and in the Results section 2.2 “Effect of repeated intraperitoneal administration of 2-AG on retinal neuroprotection and CB1R expression in the model of AMPA excitotoxicity”
Comment 6: Row 146: “AMPA induced an increase in the number of reactive microglia cells”. Present the dose and a quantitative measure, like 3 or 4 folds increase. The same for “2-AG (50μg/kg, i.p) reversed this effect”. Did it reverse the effect completely (100%)? Or only by 50 or 25%? This approach should be used for all the other assays performed in this study.
Response: We have added the x fold increase and % decrease of effect in sections of results.
Comment 7: Row 150, present the dose of AM630 that produced the effect
Response: The dose is 500μg/kg, i.p (shown in the Methods’ section and in the Figure legend).
Comment 8: Row 171, it is not clear if the assay measured all isoforms of Akt and which position is phosphoryled (ex Thr308 or Ser473). The same for ERK1 and 2.
Response: Results 2.4. “Effect of 2-AG repeated intraperitoneal administration on CB1R downstream signaling”.The following addition was made in the second paragraph.
“In this study, we employed western blot analysis using a monoclonal antibody that recognizes isoforms Akt 1, phosphorylated at the Ser473 residue. However, this antibody might also detect Akt2 and Akt3 isoforms when phosphorylated at the same residue. We also employed monoclonal antibodies that detects ERK1 (p44 MAP kinase) and ERK2 (p42 MAP kinase) when they are phosphorylated dually at Thr202/Tyr204 and Thr185/Tyr187, respectively”.
Comment 9: Row 297, there are several studies on the effect of 2AG on Akt function. Here is just one example: Type 1 cannabinoid receptor ligands display functional selectivity in a cell culture model of striatal medium spiny projection neurons. The authors should better discuss the previous data in context with their research.
Response: The following text was added to para 3 of the Discussion. “The interaction of 2-AG with Akt signaling maybe due to the CB1R dependent “biased signaling” which pertains to the ability of different cannabinoid ligands to activate divergent intracellular signaling pathways (Ibsen et al.[57]). Laprairie et al. [58] performed an extensive analysis of the functional selectivity profile of different endogenous and synthetic CB1R ligands on striatal medium spiny neurons in vitro. They discovered that the endocannabinoids 2-AG and AEA displayed a stronger biased signaling towards the Gαi/o and Gβγ-mediated Akt activation, compared to the synthetic cannabinoid ligands (Laprairie et al.[58])”.
Comment 10: There are several editing mistakes. The authors should carefully verify the paper.
Response: We edited the manuscript and hope that we have removed the mistakes.

Reviewer 2 Report
Comments and Suggestions for Authors
The study is interesting and well-presented regarding the regulation of cannabinoids receptors and their possible impact on retina and neurodegenerative retinopathies. However I have some comments which might be helpful to improve the manuscript.
Major comments
1. The results present mainly CB1R protein expression but no CB2R expression. How could authors conclude that 2-AG functioned only true CB1 and not CB2? Presenting data regarding the specificity of CB1R over CB2R would be crucial.
2. Have authors investigated other endogenous cannabinoid agonist such as 2–AGE to confirm if the observed effects are specific to 2-AG only? Testing another endogenous agonist would confirm the specificity of the presented results.
3. Have authors performed viability tests to ensure cell survival?
4. Did authors test similar experiments on human models including in vitro?
5. In the abstract, although the information mentioned regarding the experimental methods are important, including them in details in the abstract is not crucial. Instead, more information regarding 2-AG, CB1 and CB2 receptor and their contribution in ocular pathology as well as their main important results in brief would be easier to follow.
6. In the results, authors presented only protein expressions of cannabinoid receptors in their study and there is no results of gene expressions regarding the presented receptors. It would help if author present gene expressions as well. Integrating novel bioinformatic approaches will significantly improve the findings of the present research.
Minor comments
7. The introduction part could be expanded.
8. In the results section, the subheadings (2.1-2.9) describe a statement about the following paragraph but does not include the interpretation of the results in brief (a sentence). It would be more informative if they can include the final take-home message instead in their subheadings.
9. Please include negative controls for the immunofluorescence images in the figures.
10. Please include the representative fluorescent dyes (blue and red) in the figure legends.
11. There are no white arrows in Figure 8A but yellow. Please change either the color or text.
Author Response
Manuscript ID: ijms-2670262
We thank the reviewer for his/her comments.
We have revised the manuscript according to your recommendations and also responded as to why some proposed experiments were not performed.
Response to Reviewer 2
Major comments
Comment 1. The results present mainly CB1R protein expression but no CB2R expression. How could authors conclude that 2-AG functioned only true CB1 and not CB2? Presenting data regarding the specificity of CB1R over CB2R would be crucial.
Response: 2-AG activates both CB1 and CB2 receptors located on retinal neurons and glia, respectively. In the present study, we show that 2-AG, when administered i.p, activates the CB1R in retinal neurons and induces its downregulation in the 4 day model. 2-AG also activated the CB2 receptor, located mainly on microglia, as shown by the attenuation of the number of reactive microglia (Iba-1 immunoreactivity).This was supported by the data presented in this manuscript (Figure 3), showing that the CB2R antagonist AM630, but not the CB1R antagonist, reversed the 2-AG effect. Therefore, in the present study we report that 2-AG actions were mediated by both CB1 and CB2 receptors.
Comment 2. Have authors investigated other endogenous cannabinoid agonist, such as 2–AGE to confirm if the observed effects are specific to 2-AG only?
Testing another endogenous agonist would confirm the specificity of the presented results.
Response: The endocannabinoid 2-Arachidonyl glyceryl ether (AGE) is an analog of 2-AG. 2-AGE binds and activates the CB1 (Ki=21.2 ± 0.5 nM) and has very low affinity of the CB2R (Ki > 3 mM) (Hanuš et al., PNAS, 2001, Pertwee, R., Pharmacol. Therap., 1997). We did not investigate 2-AGE, but we did investigate AEA in previous studies (Kokona et al., 2015, Papadogkonaki et al., 2019). More recently we investigated the neuroprotective and anti- anti-inflammatory actions of 2-AG and the ABHD6/MAGL enzyme inhibitors in the in vivo retinal model of AMPA excitotoxicity (Kokona, Spyridakos et al. (2021). As presented in the discussion of the present manuscript there are similarities of the 2-AG actions with those of the endocannabinoid AEA, as well as synthetic cannabinoids.
Comment 3. Have authors performed viability tests to ensure cell survival?
Response: We have not performed viability tests as performed in cell cultures. However, we have shown in the Kokona, Spyridakos et al. (2021) study that 2-AG attenuated the AMPA-induced increase in TUNEL+ cells. Therefore, we did not consider necessary to perform the same experiment for the present study.
Comment 4. Did authors test similar experiments on human models including in vitro?
Response: We have not performed such a study on human retinal cell cultures. However, we were interested in investigating the mechanism involved in the 2-AG downregulation of the CB1 receptor. Papadogkonaki et al., Exp. Eye Res. (2019) was the first publication to show downregulation of the CB1 receptor in retina after subchronic/chronic treatment of cannabinoids (AEA, HU210, MethAEA).
We did perform pilot studies with 2-AG, using retinal tissues and molecular markers, important for the trafficking of GPCRs from the cell membrane to the cytoplasm, as well as the involvement of autophagy. However, the results were not favorable due to tissue limitations. In this case, cell cultures would have been a more resourceful option.
Cell cultures are the ideal means for the study of the basic mechanisms and pathways involved in the downregulation of CB1R in brain models and there are many studies in the literature regarding this question.
In the present study, we focused on the examination of CB1R expression and its response to repeated agonist stimulation in a more integrated in vivo context, which resembles the human phenotype.
Comment 5. In the abstract, although the information mentioned regarding the experimental methods are important, including them in details in the abstract is not crucial. Instead, more information regarding 2-AG, CB1 and CB2 receptor and their contribution in ocular pathology as well as their main important results in brief would be easier to follow.
Response: We agree with the reviewer. The abstract has been rewritten.
Comment 6. In the results, authors presented only protein expressions of cannabinoid receptors in their study and there are no results of gene expressions regarding the presented receptors. It would help if author present gene expressions as well. Integrating novel bioinformatic approaches will significantly improve the findings of the present research.
Response: Previous studies from our lab suggested that AEA and the synthetic cannabinoids HU210, methAEA (Papadogkonaki et al, 2019) and WIN55,212-2 (Spyridakos et al. 2021), administered intraperitoneally repeatedly led to the downregulation of the CB1 receptor. In this study, we employed real time PCR and reported that MethAEA (50μg/kg, i.p) administered for 4 days led to a significant reduction of CB1R mRNA. This suggested that the therapeutic use of these agents in chronic diseases will lead to tolerance and would not be a beneficial treatment for retinal disease.
In the present study, our major aim was to investigate whether 2-AG, administered via three different routes, repeatedly [exogenously, 2-AG, i.p. drops)/endogenously,AM11920 (MAGL/ABHD6 inhibitor),ip], would lead to CB1R downregulation.
We agree with the reviewer that performing gene expression studies and integrating novel bioinformatic approaches would improve significantly this work, but this a total new project that one cannot address at the present, and include the data in the manuscript under review.
Minor comments
Comment 7. The introduction part could be expanded.
Response: Paragraph 5 was expanded.
Comment 8. In the results section, the subheadings (2.1-2.9) describe a statement about the following paragraph but does not include the interpretation of the results in brief (a sentence). It would be more informative if they can include the final take- home message instead in their subheadings.
Response: We have added a brief statement, “take home message”, in all result sections (2.1-2.9).
Comment 9. Please include negative controls for the immunofluorescence images in the figures.
Response: Added in Methods, 4.3.2. Immunohistochemistry, 2nd para, last sentence.
Figure S3. Representative photomicrograph of retinal tissue stained only with the secondary antibody, CF543 goat anti-rabbit, as the negative control for the immunofluorescence images.
Comment 10. Please include the representative fluorescent dyes (blue and red) in the figure legends.
Response: These were added to the figure legends
Comment 11. There are no white arrows in Figure 8A but yellow. Please change either the color or text.
Response: The correction was made in the Figure Legend.

Round 2
Reviewer 2 Report
Comments and Suggestions for Authors
Thank you for your reponses.